# Efficient Long-Range Transformers: You Need to Attend More, but Not Necessarily at Every Layer

**Qingru Zhang**[†][*], **Dhananjay Ram**[◇], **Cole Hawkins**[◇], **Sheng Zha**[◇], **Tuo Zhao**[†]
[†]Georgia Institute of Technology   [◇]Amazon Web Service
{qingru.zhang,tourzhao}@gatech.edu
{radhna,colehawk,zhasheng}@amazon.com

## Abstract

Pretrained transformer models have demonstrated remarkable performance across various natural language processing tasks. These models leverage the attention mechanism to capture long- and short-range dependencies in the sequence. However, the (full) attention mechanism incurs high computational cost – quadratic in the sequence length, which is not affordable in tasks with long sequences, e.g., inputs with 8k tokens. Although sparse attention can be used to improve computational efficiency, as suggested in existing work, it has limited modeling capacity and often fails to capture complicated dependencies in long sequences. To tackle this challenge, we propose MASFormer, an easy-to-implement transformer variant with Mixed Attention Spans. Specifically, MASFormer is equipped with full attention to capture long-range dependencies, but only at a small number of layers. For the remaining layers, MASformer only employs sparse attention to capture short-range dependencies. Our experiments on natural language modeling and generation tasks show that a decoder-only MASFormer model of 1.3B parameters can achieve competitive performance to vanilla transformers with full attention while significantly reducing computational cost (up to 75%). Additionally, we investigate the effectiveness of continual training with long sequence data and how sequence length impacts downstream generation performance, which may be of independent interest.

## 1 Introduction

Pre-trained transformer models have manifested superior performance in various natural language processing tasks such as natural language modeling (NLM) (Dai et al., 2019; Radford et al., 2019), natural language generation (NLG) (Brown et al., 2020) and natural language understanding (NLU) (Devlin et al., 2019; Liu et al., 2019; He et al., 2021b).

These models leverage the attention mechanism (Vaswani et al., 2017) to compute the dependency score for each pair of tokens in an input sequence.

Some practical tasks require these transformer models to handle long-sequence inputs like 8k tokens. For example, chatbot systems gather long-term contexts of user interactions to generate informative texts (Roller et al., 2021). Summarization for news, government reports, and academic papers request models to take inputs of long sequences to generate comprehensive summaries (Shaham et al., 2022), otherwise models often miss important information. Note that typical transformer models apply full attention to capture token dependencies pair-wise. It leads to a quadratic time and space complexity w.r.t. input length. However, such a complexity is prohibitive for long sequences. In particular, it incurs massive memory consumption during the back propagation. For example, a transformer model with 250M parameters consumes over 80G GPU memory when sequence length is 8k (Zuo et al., 2022).

To address this scalability issue, various approaches have been proposed to reduce the complexity. One approach is *sparse attention*, which restricts each token to attend a subset of tokens based on predefined sparsity patterns (Beltagy et al., 2020; Zaheer et al., 2020; Ainslie et al., 2020). For instance, *block sparse attention* (Kitaev et al., 2020; Ma et al., 2023) divides the input sequence into several blocks, and only intra-block attention is performed. Besides, *sliding-window attention* (Beltagy et al., 2020; Zaheer et al., 2020; Ainslie et al., 2020) allows each token to attend to its neighboring tokens within a sliding window. These methods, though reducing the complexity of full attention, cannot sufficiently capture long-range dependencies. Other variants, such as kernel approximation (Peng et al., 2021) and low-rank approximation (Wang et al., 2020; Chen et al., 2021) methods, share the similar spirit and drawbacks. To com-

---

[*] Work was done during Qingru Zhang's internship at Amazon Web Service.

pensate for the lack of long-range dependencies, LongT5 (Guo et al., 2021) introduces global tokens that are obtained by average pooling on every block of tokens (Ainslie et al., 2020). However, the block pooling operations can weaken the signal of crucial tokens and prevent the long-range dependencies from being detected.

In addition to these methods, *state space models* (SSMs) prespecify global dependency patterns to capture the long-range dependencies only (Gu et al., 2020, 2021; Li et al., 2022; Zuo et al., 2022; Ma et al., 2023; Smith et al., 2023). These models can be regarded as linear recurrent neural networks with specifically designed fixed weights. As tailored for global dependencies, SSMs fail to effectively capture local dependencies. In order to combine both local and global dependencies, SPADE (Zuo et al., 2022) and MEGA (Ma et al., 2023) augment SSM layers into transformer layers equipped with local attention. However, state space methods require sophisticated implementation, and often encounter computational instability during the back propagation, especially when scaling up to large model size (Gupta et al., 2022). SPADE and MEGA hence inherit these drawbacks.

Note that the aforementioned methods apply same attention mechanism for every layer. We challenge this conventional wisdom and propose a transformer variant – *MASFormer* (Mixed Attention Span transFormer). MASFormer utilizes full attention only at a subset of layers whereas employs sparse attention at the remaining layers. Our design is motivated by the phenomenon – that most contexts in NLP data display a great deal of *locality of reference* (Zaheer et al., 2020; Beltagy et al., 2020). That is, most of information about a token can be derived from its neighboring tokens. In contrast, long-range dependencies among tokens are sparse and infrequent. Consider an academic paper as an example. Within a paragraph, there exist numerous short-term dependencies. Neighboring tokens are closely connected to convey meaningful semantics. Across paragraphs, there can be a small number of long-range dependencies. For example, tokens associated to the primary theme of the paper exhibit rare and weak dependencies across a long span. Since long-range dependencies occur much less frequently, a few layers of full attention are adequate to capture them. In stark contrast, short-term dependencies are more frequent, necessitating local attention in the majority of layers to fully extract

these signals.

To demonstrate the effectiveness of MASFormer, We conduct experiments on natural language modeling (ArXiv and PubMed Cohan et al. (2018)) and natural language generation (ArXiv, Cohan et al. (2018) and SCROLLS, Shaham et al. (2022)) tasks. Specifically, we compare the performance of MASFormer to other attention methods using a pretrained GPT-2 model (Radford et al., 2019) of 1.3 billion parameters. Our empirical results demonstrate that MASFormer consistently outperforms baseline methods across different attention cost (i.e. the total number of computed attention scores). In particular, MASFormer can achieve comparable performance to full attention while significantly reducing the computational cost. For example, with 27% of its attention cost, MASFormer achieves a close R2 score as full attention on QMSUM dataset.

We also make additional discoveries with MASFormer, which are of independent interest. Firstly, we investigate the effectiveness of continual training for long sequence modeling. Many publicly available models are pre-trained with sequences shorter than 2048, and often fail to perform well on longer sequences (e.g. 8k/16k tokens). To bridge the gap, we explore the option of continual training to adapt these models to long sequences, thereby avoiding pre-training from the scratch. We discuss its effectiveness with MASFormer in Section 4.3. Secondly, we showcase that increasing sequence length can yield more performance gains on downstream tasks than NLM tasks evaluated by perplexity. We are aware of the recent findings by Sun et al. (2021) that increasing context length exhibits limited impact on NLM perplexity. Nevertheless, when applying MASFormer to downstream tasks like long-context summarization, we find that model performance benefits significantly from extending context length. Such a difference arises from the fact that predicting the next tokens in NLM primarily relies on locality of reference. Capturing infrequent long-range tokens can improve perplexity but not significantly. Therefore, we emphasize the necessity to evaluate model performance on downstream tasks that require long-range dependencies. Furthermore, our empirical evidence suggests that increasing the length can improve the performance only if models possess sufficient capability to handle additional long-range information. Local attention, as a counterexample, often fails to capture long-range signals and hence

benefits much less from long sequences.

## 2 Background

### 2.1 Pretrained Language Models

Pre-trained transformer models (Devlin et al., 2019; Liu et al., 2019; Brown et al., 2020; Dosovitskiy et al., 2020; He et al., 2021b,a) have manifested superior performance in various NLP tasks. These models are often pre-trained on enormous amounts of unlabeled data in a unsupervised/self-supervised manner such that they can learn rich semantic knowledge. By further fine-tuning these pre-trained models, we can effectively transfer such knowledge to benefit downstream tasks (Zhang et al., 2023).

Existing research on long-range transformers commonly requires pre-training the proposed models from scratch to accommodate new architectures and long inputs (Guo et al., 2021; Zuo et al., 2022). However, the significant training overheads raise a barrier for the widespread utilization of these methods across different language models. Motivated by this, we explore the possibility of leveraging existing pre-trained models and adapting them to long sequences though continual training.

### 2.2 Attention Mechanism

Suppose the input to the layer is $X \in \mathbb{R}^{n \times d}$, where $n$ is the input sequence length and $d$ is embedding dimension, then self-attention mechanism outputs

$$\text{Attn}(X) = \text{softmax}\left(\frac{QK^\top}{\sqrt{d}}\right)V \qquad (1)$$

where $Q = XW_q, K = XW_k, V = VW_v$ and $W_q, W_k, W_v$ are learnable projection weights. Such full attention can simultaneously evaluate the alignment between any pair of tokens in the sequence. Specifically, denote the attention score $A = \text{softmax}(QK^\top/\sqrt{d})$, then $A_{ij}$ captures the alignment between tokens $i$ and $j$. A typical transformer model applies the full attention at every layer. Denote the number of layers as $L$. Then its attention cost is $Ln^2$.

Sparse attention variants are introduced to mitigate the computational cost of full attention. Figures 1a and 1b illustrates the attention patterns of block sparse attention and sliding-window attention. For instance, block sparse attention divides tokens into blocks of size $b$ and performs intra-block attention only, resulting in an attention cost of $bn$. Sliding-window attention allows each token to attend its left/right neighboring tokens within a local window of size $w$. In most of cases, block sparse attention exhibits similar performance as sliding-window attention (Zuo et al., 2022).

## 3 Our Approach

We present our method – MASFormer, a long-range transformer variant that mixes different attention spans across layers.

### 3.1 MASFormer: Mixed Attention Span

MASFormer leverages full attention exclusively at a subset of transformer layers, whereas it employs block sparse attention at the remaining layers. The structure of MASFormer is illustrated in Figure 1c. We choose full attention to encode long-range information due to the following reasons: (i) full attention exhibits superior capability to capture long-range dependencies compared to sparse attention; (ii) full attention does not require sophisticated implementation and hence is computationally stable compared to SSMs (Zuo et al., 2022; Gupta et al., 2022); (iii) full attention is compatible with existing pre-trained transformer models, enabling us to conduct continual training which we elaborate in Section 3.2. To mitigate the computational cost, we restrict the number of layers using full attention.

MASFormer is motivated by empirical investigations on performance comparison between models that apply the same attention span at every layer. Figure 2 presents the performance of block sparse attention and full attention on language modeling and summarization tasks. We find that, given long-sequence inputs, sparse attention is often insufficient to capture long-range dependencies beyond its attention span. As a result, it shows unsatisfactory performance. To remedy it, one can either increase attention span or switch to full attention to improve model capability of capturing sophisticated dependencies. Though improving model performance, it incurs high computational cost.

Confronting such a trade-off between computational cost and model performance, we challenge the common practice – *that applies the same attention span at every layer.* MASFormer provides an alternative solution. Instead of increasing attention span evenly, MASFormer allocates a large portion of attention computations to a subset of $l$ layers by equipping them with full attention. Specifically, equipping bottom layers with full attention can yield the best performance as suggested by our empirical analysis in Section 4.3[1]. At the remain-

---

[1]Please see Section 4.3 for detailed explanations

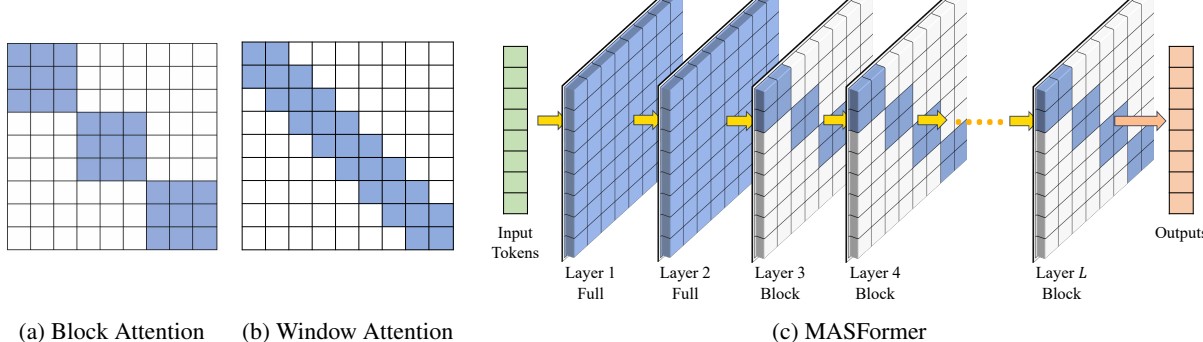

(a) Block Attention      (b) Window Attention           (c) MASFormer

Figure 1: Illustration of attention patterns of (a) block sparse attention with block size $b = 3$; (b) sliding-window attention with window size $w = 1$ (on each side); (c) MASFormer that integrates full and sparse attention.

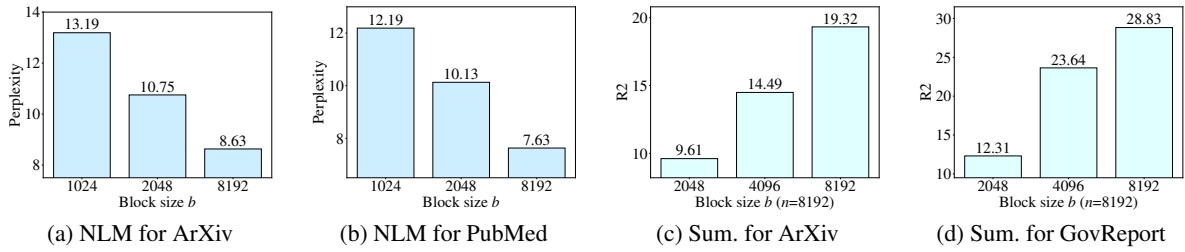

(a) NLM for ArXiv     (b) NLM for PubMed     (c) Sum. for ArXiv     (d) Sum. for GovReport

Figure 2: (a,b): We evaluate the perplexity of a pre-trained GPT-2 model with block attention of differnet block size after continual training. (c,d): We fine-tune a GPT-2 model with block attention and compare the summarization performance on ArXiv and GovReport under different block size. Here the input length $n$ is 8192.

ing layers, MASFormer utilizes block attention of small size $m$, resulting in a controlled attention cost of $(L - l)mn + ln^2$. As mentioned in Section 1, such a design is inspired by the phenomenon that most of contexts in NLP data exhibit a great deal of locality of reference. Long-range dependencies, in contrast, are less frequent. Therefore, it is not necessary to enhance attention span at every layer. Instead, a few layers of full attention are sufficient to capture infrequent long-range signals. The majority of layers can maintain small attention spans to adequately extract local dependencies and control the attention cost.

Our empirical results demonstrate that, with the same attention cost, MASFormer significantly outperforms sparse attention. Remarkably, MAS-Former can achieve comparable performance to full attention while substantially reducing computational cost. Therefore, by mixing different attention spans, MASFormer strikes a better balance between computational cost and model performance.

Moreover, MASFormer offers additional implementation advantages. As using the same attention function, MASFormer is easy to implement and compatible with existing pre-trained models. We can build MASFormer upon pre-trained transformers by changing their attention patterns, which does not involve modification on model architectures

and pre-trained weights. Meanwhile, acceleration packages, such as FlashAttention (Dao et al., 2022) and xFormers (Lefaudeux et al., 2022), are applicable to further accelerate the computation of block attention and full attention in MASFormer.

## 3.2 Continual Training with Long Sequences

As mentioned, MASFormer can be implemented upon majority of pre-trained transformers by modifying their attention patterns. However, most of publicly available models are pre-trained with sequences shorter than 2048, and often exhibit subpar performance on longer sequences such as 8k/16k. To bridge this gap, we propose the continual training to adapt the revised model on long sequences and new attention pattern. As such, we can preserve existing pre-trained knowledge and circumvent the intensive overheads of pre-training from scratch. In particular, we first modify the attention pattern of the target model as proposed by MASFormer. If the pre-trained model uses absolute position embeddings, we duplicate them to accommodate long sequences. Subsequently, we provide the revised model with long sequences (e.g., 8k) from pre-training corpus like PILE (Gao et al., 2020). Then we conduct continual pre-training using casual language modeling (CLM) objective. We discuss the effectiveness of continual training in Section 4.3.

# 4 Experiments

We evaluate the effectiveness and efficiency of MASFormer on natural language modeling (ArXiv and PubMed, Cohan et al. (2018)), natural language generation (ArXiv Cohan et al. (2018), QMSUM and GovReport Shaham et al. (2022)). We choose the GPT-3 XL model architecture (Brown et al., 2020) as our base model, which consists of 1.3 billion parameters and 24 layers and is pre-trained on PILE (Gao et al., 2020) for 300 billion tokens. GPT is a general purpose model that can be applied to many tasks instead of tailoring them for specific tasks. As such, it makes easy to control experiments and showcase the difference among various methods.

**Implementation Details.** Our base model uses absolute positional embeddings with maximum length 1024. To accommodate longer inputs, we duplicate its positional embeddings to have the maximum length as 8192 such that the model can handle sequences containing up to 8192 tokens. Then, we implement different attention methods by modifying the attention pattern of the base model. We implement all the models with *PyTorch* (Paszke et al., 2019). All the experiments are conducted on NVIDIA A100 GPUs.

**Continual Training Details.** After changing the attention pattern, we conduct the continual training for MASFormer and baseline methods on PILE corpus (Gao et al., 2020) to adapt the revised models to new attention patterns and long-sequence inputs. We leverage the casual language modeling (CLM) objective to train the model for 50,000 steps with a warmup of 2000 steps. We set the input length as 8192 and use a batch size of 128 such that the models are optimized with 1M tokens per step. We use the constant learning 0.0001 for all methods.

**Baseline.** We compare MASFormer with the following methods:

• *All full attention* is to apply full attention at every layer. It has been adopted by most of existing transformer models as default. Although incurring the maximum attention cost, it achieves the best performance for most of our tasks. Hence, it acts as an upper bound for other methods.

• *All block sparse attention* is to apply block attention at every layer, which is an effective method to reduce computational cost when modeling long sequences. Block attention sets the attention span of each layer identical such that it evenly distributes the budget of attention computation across layers.

• *All sliding-window attention* is to apply sliding-window attention at every layer, which is another variant of sparse attention. It shares the similar spirits and often performs similarly as block attention.

In the following experiments, we compare MASFormer and the baseline methods across different attention cost $\mathcal{C}$. That is, for all block sparse attention, we set the block size as $b = \mathcal{C}/(Ln)$. For all sliding-window attention, we choose the window size as $w = \mathcal{C}/(2Ln)$. For MASFormer, we apply a small block size $m = 1024$ for its block attention and set $l$ as $(\mathcal{C} - Lmn)/(n^2 - mn)$. Then we observe how their performance evolves when enhancing the attention cost $\mathcal{C}$ or input length $n$.

**Experiment Overview.** We briefly summarize the experimental contents as follows:

• Section 4.1 presents the perplexity evaluation of all the models on ArXiv and PubMed after continual training.

• Section 4.2 compares the summarization performance of the models on ArXiv, QMSUM, and GovReport after fine-tuning. Besides, we also discuss the difference between perplexity and downstream evaluation in reflecting model capacity to capture long-range dependencies.

• Section 4.3 provides three crucial analyses: (i) we evaluate the benefits of increasing input length and discuss the requirements to attain these gains; (ii) we analyze the effectiveness of continual training for long-sequence modeling; (iii) we conduct an ablation study to demonstrate that equipping *bottom layers* with full attention yields the most significant performance gains than other options. We further provide the explanations.

## 4.1 Natural Language Modeling

### 4.1.1 Datasets and Evaluation Details

**Datasets.** We evaluate the perplexity of the updated GPT-2 for each attention method after continual training. The evaluation is conducted on test sets of ArXiv and PubMed (Cohan et al., 2018). Table 5 presents the statistics of these two datasets. Pubmed consists of scientific documents, with a document's content used as input and its corresponding abstract as the target summary. ArXiv is similar to PubMed, with documents from arXiv.

**Evaluation Details.** We conduct the perplexity evaluation under two settings. (i) We calculate the perplexity (ppl.) with all documents from test sets. Table 1 presents the overall perplexity of different models on two datasets. (ii) To showcase

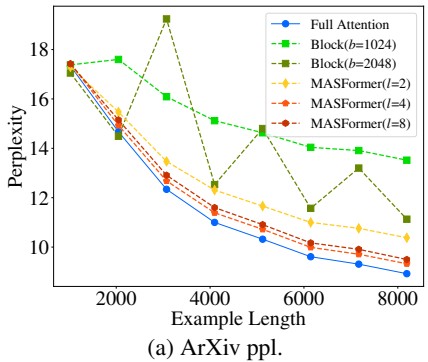 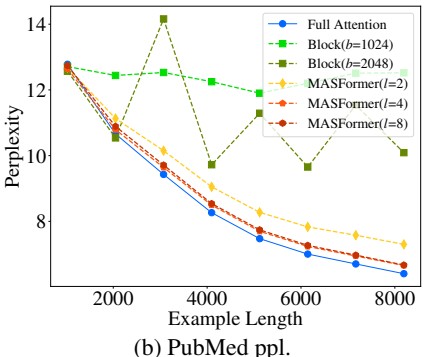

(a) ArXiv ppl.  (b) PubMed ppl.

Figure 3: Perplexity evaluation on ArXiv and PubMed with examples of different length. Here $x$-axis is the maximum document length of each subset, i.e., $k \times 1024$ ($k = 1, 2, 3, \dots$).

the varying behaviors of models on documents of different length, we divide all documents into several subsets according to their length. Each subset consists of examples, whose length is within $((k-1) \times 1024, k \times 1024]$ ($k = 1, 2, 3, \dots$). Then, we evaluate the perplexity on each subset. Figure 3 presents the perplexity of models on different subsets of examples.

### 4.1.2 Results

Table 1 compares the overall perplexity on test sets of ArXiv and PubMed. The results suggest that, with $l = 4$ layers of full attention, MASFormer achieves comparable performance to all full attention, while reducing 72% of its attention cost. With the similar attention cost $\mathcal{C}$, MASFormer outperforms all block attention that evenly distributes the budget of attention computation. For example, MASFormer with $l = 2$ achieves 8.75 ppl. on PubMed, which is 1.37 lower than that of block attention of $b = 2048$.

| Methods | $\mathcal{C}$ | ArXiv | PubMed |
|---|---|---|---|
| Full attention | 1,610M | 8.63 | 7.63 |
| Block ($b$=1024) | 201M | 13.19 | 12.19 |
| Block ($b$=2048) | 402M | 10.75 | 10.13 |
| MASFormer ($l$=2) | 318M | 10.25 | 8.75 |
| MASFormer ($l$=4) | 436M | 9.31 | 8.25 |
| MASFormer ($l$=8) | 671M | 9.63 | 8.25 |

Table 1: Perplexity evaluation on ArXiv and PubMed.

Figure 3 illustrates the perplexity variation of each method given examples of different length. We can tell that MASFormer and full attention show better performance on longer documents, suggesting increasing context length can improve their prediction performance. Full attention, though incurring the highest attention cost, always achieves the best performance due to its outstanding capability to handle sophisticated dependencies. Notably,

with 27% of its attention cost, MASFormer exhibits a curve of ppl. v.s. length that closely resembles to that of full attention. This demonstrates the effectiveness and efficiency of MASFormer to capture long-range dependencies. In contrast, block sparse attention benefits much less from long contexts and underperforms both of them because of its incapability to encode long-range signals. For example, when $b = 1024$, block attention achieves similar perplexity on PubMed examples of different length.

## 4.2 Natural Language Generation

### 4.2.1 Datasets and Training Details

**Datasets.** We evaluate the downstream performance of models on several abstractive summarization tasks to compare their capability of handling long sequences in practice. Specifically, we fine-tune models on ArXiv (Cohan et al., 2018), QM-SUM and GovReport (from SCROLLS benchmark, Shaham et al. (2022)). Their statistics are summarized in Table 5. We mainly use ROUGE-2 (R2) score (Lin, 2004) as the evaluation metric, which is more important and sensitive than R1 and RL.

**Training Details.** After continual training, we fine-tune each model and report R2 scores on validation sets. Specifically, we fine-tune models for 3000 steps on QMSUM, 8000 steps on GovReport, and 12000 steps on ArXiv. We set the batch size as 64 for ArXiv and 32 for QMSUM and GovReport. We pick the learning rates from $\{1 \times 10^{-5}, 5 \times 10^{-5}, 1 \times 10^{-4}, 5 \times 10^{-4}\}$, and choose the optimal ones to report the performance of each method. Moreover, the input length is fixed as 8192. We apply the greedy decoding for generation. Please see Appendix B for more details.

### 4.2.2 Results

In Table 2[2] and Figure 4, we present the fine-tuning results on QMSUM, ArXiv and GovReport across

---

[2]Please see Table 6 in Appendix B for all ROUGE scores.

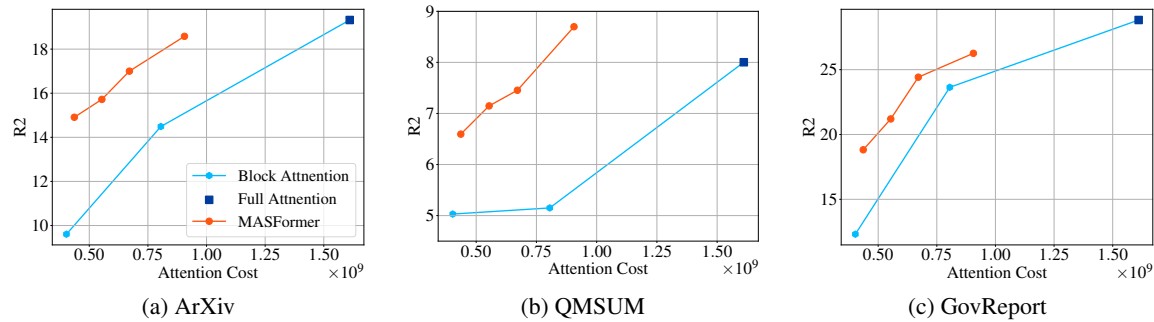

Figure 4: Given input length as 8192, we compare summarization performance between MASFormer and block/full attention when increasing the attention cost.

| Methods | $\mathcal{C}$ | QMSUM | ArXiv | GovReport |
|---|---|---|---|---|
| Full attention | 1610M | 8.00 | 19.32 | 28.83 |
| Window($w$=1024) | 402M | 4.32 | 13.51 | 17.03 |
| Block($b$=2048) | 402M | 5.03 | 9.61 | 12.31 |
| MASFormer($l$=4) | 436M | **6.59** | **14.91** | **18.82** |
| Window($w$=2048) | 805M | 5.05 | 15.21 | 22.79 |
| Block($b$=4096) | 805M | 5.15 | 14.50 | 23.64 |
| MASFormer($l$=6) | 553M | 7.15 | 15.72 | 21.20 |
| MASFormer($l$=8) | 671M | **7.46** | **17.00** | **24.42** |
| MASFormer($l$=12) | 906M | 8.70 | 18.58 | 26.26 |

Table 2: Summarization performance of models with different attention methods. The best results are shown in **bold**.

different attention cost. The results demonstrate that, with the similar attention cost, MASFormer significantly outperforms sparse attention variants. Furthermore, when enhancing attention cost, MAS-Former achieves greater performance gains than sparse attention methods. This is evident from the steeper slope of its R2 curve versus attention cost, in contrast to the baseline method. For example, when increasing $\mathcal{C}$ form 553M to 671M, the R2 score of MASFormer on QMSUM exhibits a substantial improvement, reaching 8.70 from 7.46. Remarkably, this score surpasses even that of full attention. Therefore, MASFormer addresses the trade-off between computational cost and performance gains in a more efficient and effective way.

Notice that, in order to achieve comparable summarization performance to full attention, MAS-Former needs at leaset $l = 8$ layers of full attention, and providing more can lead to more gains. This observation is different from the findings in NLM (Figure 3) that increasing $l$ beyond 4 provides limited improvement in perplexity. Their different capacity requirements arise from the fact that predicting next tokens in NLM primarily relies on lo-

cal dependencies. Capturing infrequent long-range tokens does not significantly improve perplexity. Thus, this discrepancy emphasizes the necessity to evaluate long-range models on downstream tasks.

### 4.3 Analysis

#### 4.3.1 Benefits of Increasing Sequence Length

In this section, we investigate the benefits of increasing input length for downstream performance. Specifically, we select the input length from $\{2048, 4096, 6144, 8192\}$ and present the fine-tuning performance of full attention in Figure 5. The results consistently demonstrate that as the input length increases, the model's performance improves. That is, downstream performance benefits significantly from long-sequence inputs. In contrast, increasing example length beyond 6k results in marginal improvements in perplexity (See Figure 3), highlighting again the importance of downstream evaluation.

In addition, when comparing the behaviors of block attention in Figure 2c and 2d, we find that sparse attention often insufficiently capitalize on the benefits offered by longer inputs. For instance, given block size as 4096, its performance on ArXiv remains nearly unchanged when increasing input length from 4096 (R2 $= 15.52$ in Figure 5a) to 8192 (R2 $= 14.49$ in Figure 2c). This finding suggests that enhancing input length can only improve model performance if the model possesses the sufficient capability to handle long-range dependencies.

#### 4.3.2 Effectiveness of Continual Training

We analyze the effectiveness of continual training by comparing fine-tuning performance of MAS-Former ($l = 8$) under the following settings: (i) MASFormer without continual training (w.o. C.T.); (ii) MASFormer continually trained with short inputs (C.T. ($n$=2048)); (iii) MASFormer continually trained with long inputs (C.T. ($n$=8192)). Table 3

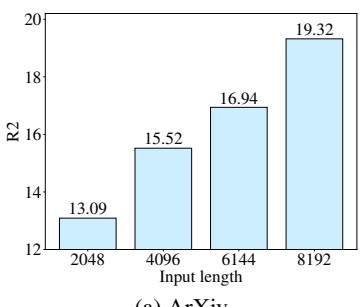 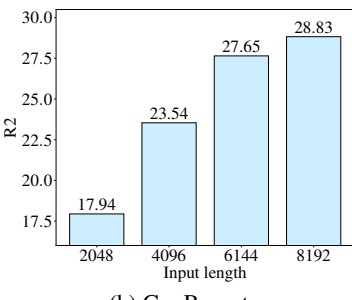 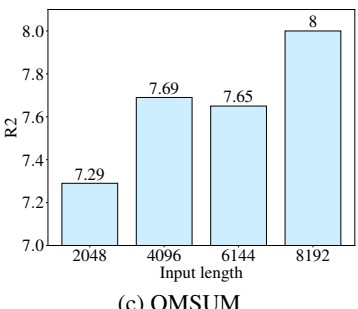

| (a) ArXiv | (b) GovReport | (c) QMSUM |

Figure 5: Fine-tuning performance of full attention under different input length.

presents fine-tuning performance of these models. We can tell that continual training with long inputs indeed facilitates the revised models to adapt to new structures and long-sequence inputs.

| $l = 8$ | QMSUM | GovReport |
|---------|-------|-----------|
| w.o. C.T. | 29.33/6.43/25.71 | 53.28/23.61/51.74 |
| C.T. ($n$=2048) | 29.87/7.16/26.15 | 52.28/23.01/49.83 |
| C.T. ($n$=8192) | **30.91/7.45/27.02** | **54.37/24.42/51.87** |

Table 3: We report R1/R2/RL for the above results.

### 4.3.3 Where to use full attention

To answer where to apply full attention, we compare fine-tuning performance of MASFormers that apply full attention at (i) bottom layers; (ii) middle layers; (iii) top layers; (iv) every $L/l$ layers. The results in Table 4 demonstrate that equipping bottom layers with full attention yields the best performance. This is because that long-range dependencies can be continually captured and reinforced by bottom layers before propagated to upper layers. As such, these long-range signals can be effectively incorporated into the upper layers with local attention, facilitating their encoding of local information. In contrast, when equipping local attention at bottom layers, long-range tokens are first aggregated with neighboring tokens by local attention, thereby weakening their long-range signals. Moreover, if alternating full and local attention every $L/l$ layers, the long-range signals cannot be continually reinforced nor efficiently captured.

## 5 Discussion

GPT-Neo (Black et al., 2021) introduces an attention pattern that alternates full and window attention. However, this models is not tailored for long sequences. It sets the local window size as 256 and has the maximum input length as 2048, unable to handle long sequences. Instead, this attention pattern is applied heuristically in an attempt to re-

| Position | QMSUM | GovReport |
|----------|-------|-----------|
| Every 3 | 28.26/6.94/25.03 | 26.16/12.37/24.82 |
| Top 8 | 20.89/4.52/18.37 | -/-/- |
| Middle 8 | 27.27/5.99/24.06 | 20.80/9.01/19.52 |
| Bottom 8 | **30.91/7.45/27.02** | **54.37/24.42/51.87** |
| Every 2 | 31.27/8.19/27.41 | 35.34/16.04/33.68 |
| Bottom 12 | **32.53/8.70/28.75** | **56.98/26.26/54.46** |

Table 4: Performance comparison of MASFormers that apply full attention at different layers (# layers $L$=24).

duce computational cost. However, as discussed in Section 4.3.3, this approach is neither effective nor efficient as MASFormer when handling long sequences. As shown in Table 4, applying full attention at every 2 or 3 layers underperforms applying it at bottom 12 or 8 layers. Therefore, alternating between full and block attention results in additional computational cost and performance degradation.

In contrast, MASFormer presents an effective solution for efficient long-sequence modeling. It provides guidance on adapting existing pre-trained transformers to long inputs. Meanwhile, it provides insights for designing large long-range models, especially for deeper models. By equipping only a subset of bottom layer with full attention, we can substantially mitigate computational cost. Additionaly, the computation of MASFormer can be further optimized by leveraging system-level acceleration techniques (e.g., FlashAttention and xFormer) that support both block and full attention.

## 6 Conclusion

We propose an efficient long-range transformer – MASFormer that utilizes full attention at a few of bottom layers and employs sparse attention at the remaining layers. Our empirical results on natural language modeling and generation tasks demonstrate that MASFormer can address the trade-off between computational cost and performance gains in a more efficient and effective way.

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

## A Dataset Statistics

In the following table, we provide the detailed statistics of datasets in our experiments, including example splits and length statistics.

| Dataset | Example Count | | | Input Length | | |
|---|---|---|---|---|---|---|
| | Train | Valid | Test | Average | Median | 90th percentile |
| ArXiv | 203,037 | 6,436 | 6,440 | 10,720 | 8,519 | 20,170 |
| PubMed | 119,924 | 6,633 | 6,658 | 4,748 | 3,883 | 8,883 |
| QMSUM | 1,257 | 272 | 281 | 9,497 | 14,197 | 27,761 |
| GovReport | 17,457 | 972 | 973 | 7,886 | 8,841 | 18,835 |

Table 5: Statistics of datasets. Input length measured in tokens using a SentencePiece Model.

## B Natural Language Generation

### B.1 The Results of All ROUGE Scores

| Methods | $\mathcal{C}$ | QMSUM | ArXiv | GovReport |
|---|---|---|---|---|
| Full attention | 1610M | 31.50 / 8.00 / 27.81 | 46.13 / 19.32 / 41.89 | 60.53 / 28.83 / 57.88 |
| Window ($w$=1024) | 402M | 23.31 / 4.32 / 20.62 | 35.90 / 13.51 / 32.19 | 49.82 / 17.03 / 47.42 |
| Window ($w$=2048) | 805M | 26.73 / 5.05 / 23.40 | 38.74 / 15.21 / 34.87 | 56.14 / 22.79 / 53.50 |
| Block ($b$=2048) | 402M | 26.24 / 5.03 / 23.13 | 21.85 / 9.61 / 19.86 | 26.37 / 12.31 / 25.18 |
| Block ($b$=4096) | 805M | 26.96 / 5.15 / 23.85 | 35.95 / 14.50 / 32.37 | 49.83 / 23.64 / 47.50 |
| MASFormer ($l$=4) | 436M | 29.86 / 6.59 / 25.87 | 38.85 / 14.91 / 34.98 | 46.67 / 18.82 / 44.39 |
| MASFormer ($l$=6) | 553M | 30.83 / 7.15 / 27.12 | 36.29 / 15.72 / 32.96 | 49.26 / 21.20 / 46.89 |
| MASFormer ($l$=8) | 671M | 30.91 / 8.00 / 27.81 | 43.31 / 17.00 / 39.12 | 54.37 / 24.42 / 51.87 |
| MASFormer ($l$=12) | 906M | 32.53 / 8.70 / 28.75 | 45.19 / 18.58 / 40.72 | 56.98 / 26.26 / 54.46 |

Table 6: Finetuning performance of different attention methods.

### B.2 Training Details

| Methods | QMSUM | ArXiv | GovReport |
|---|---|---|---|
| Full attention | $1 \times 10^{-5}$ | $1 \times 10^{-4}$ | $1 \times 10^{-4}$ |
| Window attention ($w$=1024) | $5 \times 10^{-4}$ | $5 \times 10^{-5}$ | $5 \times 10^{-4}$ |
| Window attention ($w$=2048) | $5 \times 10^{-5}$ | $5 \times 10^{-5}$ | $1 \times 10^{-4}$ |
| Block attention ($w$=2048) | $1 \times 10^{-4}$ | $1 \times 10^{-5}$ | $1 \times 10^{-5}$ |
| Block attention ($w$=4096) | $5 \times 10^{-5}$ | $5 \times 10^{-4}$ | $1 \times 10^{-5}$ |
| MASFormer ($l$=4) | $5 \times 10^{-5}$ | $1 \times 10^{-3}$ | $5 \times 10^{-4}$ |
| MASFormer ($l$=6) | $5 \times 10^{-5}$ | $5 \times 10^{-5}$ | $5 \times 10^{-4}$ |
| MASFormer ($l$=8) | $5 \times 10^{-5}$ | $5 \times 10^{-4}$ | $5 \times 10^{-4}$ |
| MASFormer ($l$=12) | $1 \times 10^{-5}$ | $1 \times 10^{-4}$ | $1 \times 10^{-4}$ |

Table 7: The fine-tuning learning rate of each method on each dataset.

We conduct continual training for all attention methods with training date form PILE and input length as 8192. After continual training, we obtain the continually trained models for each method and fine-tune them on QMSUM, ArXiv and GovReport to compare their summarization performance. During the

fine-tuning, we set the input length as 8192 for all datasets and all models. We apply the greedy decoding for generation and set the maximum output length as 256 for QMSUM, 1024 for GovReport, and 512 for ArXiv. Table 8 lists the details of these hyperparameters. Besides, we apply the linear learning rate schedule to fine-tune the models and the base learning rates are summarized in Table 7.

| Hyperparameter | QMSUM | ArXiv | GovReport |
|---|---|---|---|
| Training steps | 3000 | 12000 | 8000 |
| Batch size | 32 | 32 | 64 |
| Input length | 8192 | 8192 | 8192 |
| Maximum generation length | 256 | 512 | 1024 |
| Weight decay | 0.001 | 0.001 | 0.001 |

Table 8: The other fine-tuning parameters for each dataset, which remain the same for every method.