# OpenReview forum: "Efficient Long-Range Transformers: You Need to Attend More, but Not Necessarily at Every Layer"
_EMNLP/2023/Conference — EMNLP 2023 Findings_

### Official Review · Reviewer_THaz · 2023-08-02

**Typos Grammar Style And Presentation Improvements:** line 461
**Soundness:** 4

**Excitement:**

3: Ambivalent: It has merits (e.g., it reports state-of-the-art results, the idea is nice), but there are key weaknesses (e.g., it describes incremental work), and it can significantly benefit from another round of revision. However, I won't object to accepting it if my co-reviewers champion it.

**Paper Topic And Main Contributions:**

Authors observe that for long-range language modeling and summarization tasks, it is more important to have layers that can capture long-range dependencies in the document early in the computation than later. Given a pre-trained Transformer LM, they propose to use chunked attention with non-overlapping chunks at all but bottom few layers where they retain full attention. They show that this saves compute while maintaining performance on long-range language modeling and summarization tasks. They show that sparsifying alternate layers, middle, bottom layers is not as effective as sparsifying the top layers.

**Questions For The Authors:**

A. Table 1 : perplexity of sliding window is missing. Block atention cannot exchange info accross blocks - but allowing neighbouring blocks to communicate can alow info to propagate as we go up the layers. So sliding window ppl needs to be evaluated.

**Reasons To Accept:**

1. The problem is well motivated and the proposed trick is very simple, practical and can be used plug-and-play with the most commonly used LMs. Importantly, utilizing Flash attention, etc is also easy which is big plus.
2. Experiments are sensible and results are fair - they do save compute and the degradations are not too significant.

While the paper is not suitable for main conference, I do think that these findings are interesting enough for *Findings track*.

**Reasons To Reject:**

There is no novelity - no new techniques are introduced.

**Reproducibility:**

5: Could easily reproduce the results.

**Reviewer Confidence:**

5: Positive that my evaluation is correct. I read the paper very carefully and I am very familiar with related work.

---

> ### Author Rebuttal · Authors · 2023-08-28
>
> We appreciate the thoughtful review and constructive feedback from the reviewer. In response to the reviewer's comments, we provide the following clarifications and explanations.
>
> **Q1: The novelty of our method.**
>
> We respectfully disagree with the reviewer’s view on the novelty of our approach. It is acknowledged that sparse attention and continual training have been respectively explored in the prior literature. Nevertheless, we are the first to apply the mixed attention span and continual training to long sequence modeling and combine them to resolve a new question. That is how to construct efficient long-range transformers based on well-established pre-trained models. This is our key novelty. The findings gained through MASFormer provide a deeper understanding of long-sequence modeling. Our method appears straightforward but is non-trivial. We value simplicity over unnecessary complexity: if the question can be effectively resolved by a simple approach, additional complexities may not be necessary. In this light, we would advocate our approach for its practical effectiveness, implementation friendliness, and compatibility with existing pre-training progress.
>
> **Q2: The perplexity results of sliding window attention.**
>
> Thanks for your helpful suggestions. We completed the perplexity evaluation for sliding window attention. Similar as Figure 3, the following tables present the comparison among methods with close attention cost.
>
> Table 1: The perplexity evaluation on examples of PubMed.
> | Methods \ Example length | 4096 | 5120 | 6144 | 7168 | 8192 |
> | :-----: | :-----: | :-----: | :-----: | :-----: | :-----: |
> | Full attention | 8.26 | 7.47 | 7.00 | 6.70 | 6.41 |
> | Window ($w$=1024) | 8.06 | 7.67 | 7.89 | 8.21 | 8.17 |
> | Block ($b$=2048) | 9.72 | 11.29 | 9.65 | 11.54 | 10.09 |
> | MASFormer ($l$=4) | 8.49 | 7.69 | 7.23 | 6.94 | 6.65 |
>
> Table 2: The perplexity evaluation on examples of ArXiv.
> | Methods \ Example length | 4096 | 5120 | 6144 | 7168 | 8192 |
> | :-----: | :-----: | :-----: | :-----: | :-----: | :-----: |
> | Full attention | 11.00 | 10.32 | 9.61 | 9.31 | 8.92 |
> | Window ($w$=1024) | 10.25 | 9.80 | 9.29 | 9.14 | 8.88 |
> | Block ($b$=2048) | 12.53 | 14.80 | 11.57 | 13.20 | 11.13 |
> | MASFormer ($l$=4) | 11.39 | 10.72 | 9.99 | 9.71 | 9.32 |
>
> From the results, we can tell that MASFormer surpasses both block and window attentions when handling long-sequence examples (>6k) on PubMed. On the ArXiv dataset, window attention exhibits the best performance. Generally, window attention achieves better performance than block attention due to the reason mentioned by the reviewer – the information can propagate among windows when going forward to upper layers. As a result, multi-layer window attention can capture longer local dependencies than block attention, resulting in better perplexity.
>
> As discussed in Section 1(line 172), however, the perplexity performance primarily relies on capturing local dependencies and hence cannot practically reflect the downstream performance. When evaluated on downstream summarization tasks, window attention consistently underperforms MASFromer and full attention (as demonstrated in Table 2). Therefore, window attention is adept at capturing local dependencies but ineffective in long-range dependencies.
>
> We kindly remind the reviewer that sliding window attention can also be integrated into our approach, replacing the block attention. The reason of selecting the block attention for MASFormer is that its better compatibility with FlashAttention provides additional acceleration: FlashAttention actually provides a function dedicated to the block attention, leading to a better speedup than the sliding window attention (please see FlashBlocksparseAttention in the official flash-attention repo). Note that, if computationally affordable, the sliding window attention can also be used in MASFormer, which has the potential to further improve the performance. However, it takes lots of time and requires substantial computational resources to re-train new MASFormers with the sliding window attention. Unfortunately, our access to computational resources has been constrained due to adjusted priorities after the EMNLP submission. We are committed to incorporating additional experiments with sliding window attention in our next version.

---

### Official Review · Reviewer_xgsE · 2023-08-03

**Soundness:** 4

**Excitement:**

4: Strong: This paper deepens the understanding of some phenomenon or lowers the barriers to an existing research direction.

**Paper Topic And Main Contributions:**

In this paper the authors propose a new efficient long-range transformer, the MASFormer (Mixed Attention Span Transformer).

Motivated by the fact that most of the relevant information about a token can be derived from its neighboring tokens and that long-range dependencies are sparse and infrequent, the authors propose a transformer with some layers of full-attention and some layers with block sparse attention.

Also, instead of training the model, the authors simply continue gpt2’s training using bigger sequence lengths and the proposed attention.

The authors perform experiments on language modeling and summarization, which showcase the advantages provided by MASFormer.


**Questions For The Authors:**

- Why haven’t you included an analysis of the time and space effectively required for each model?


**Reasons To Accept:**

- The paper is clear and well written.
- The model proposed is quite simple and performs well when compared with the vanilla transformer while being more efficient. With similar attention costs, MASFormer performs better than transformers with block sparse attention or sliding window attention in all layers.
- The method works well on top of a pre-trained model. Saving the time to pre-train the model from scratch.
- The authors provide a detailed analysis, which shows that having the full-attention layers at the bottom of the transformer works best and that continuing the model training with the proposed attention mechanisms and higher sequence length improves the downstream performance.


**Reasons To Reject:**

- The paper is missing a comparison between the time and space effectively required for each model.


**Reproducibility:**

4: Could mostly reproduce the results, but there may be some variation because of sample variance or minor variations in their interpretation of the protocol or method.

**Reviewer Confidence:**

4: Quite sure. I tried to check the important points carefully. It's unlikely, though conceivable, that I missed something that should affect my ratings.

---

> ### Author Rebuttal · Authors · 2023-08-28
>
> We appreciate the thoughtful review and constructive feedback from the reviewer. We sincerely thank the reviewer for acknowledging the effectiveness and efficiency of our method. We provide the following response to the reviewer’s question.
>
> **Q: The analysis of the time and space effectively required for each model.**
>
> Thanks for your helpful suggestion. We kindly remind the reviewer that the time and space complexities are closely tied to the attention cost $\mathcal{C}$, which is the practical cost of attention computations. In this regard, memory footprint and computational speed roughly align with the attention cost. We compare different methods under the same attention cost. While incurring similar time and space complexity, MASFormers consistently achieve much better performance than block/window attention baselines (please see Figure 4 and Table 2).
>
> Notice that, the practical training acceleration primarily depends on intricate implementation details. For example, we can employ a function of FlashAttention dedicated to block attention to accelerate the computation of block attention (please see FlashBlocksparseAttention in the official flash-attention repo). In contrast, window attention is not compatible with this function and hence exhibits less speedup.  In practice, when fixing attention cost as, e.g.,~430M and sequence length as 8k on summarization tasks, MASFormer and block attention show similar memory footprint and training speed which is 11% faster than the window attention. In the same setting, they can save 35% memory consumption and achieve ~45% training speedup compared to full attention.

---

### Official Review · Reviewer_F2Z8 · 2023-08-12

**Typos Grammar Style And Presentation Improvements:** None
**Soundness:** 4

**Excitement:**

4: Strong: This paper deepens the understanding of some phenomenon or lowers the barriers to an existing research direction.

**Missing References:**

None

**Paper Topic And Main Contributions:**

The authors tried to lower the quadratic complexity of the self-attention mechanism by proposing MASFormer, which has self-attention in very few layers and sparse attention in the rest of the layers.

**Questions For The Authors:**

None

**Reasons To Accept:**

The new architecture MASFormer effectively combines self-attention and sparse attention mechanisms in a transformer network. By doing this the authors can significantly reduce the computation cost without impacting too much on the performance.

**Reasons To Reject:**

None

**Reproducibility:**

5: Could easily reproduce the results.

**Reviewer Confidence:**

3: Pretty sure, but there's a chance I missed something. Although I have a good feel for this area in general, I did not carefully check the paper's details, e.g., the math, experimental design, or novelty.

---

> ### Author Rebuttal · Authors · 2023-08-28
>
> We appreciate the helpful feedback and thank the reviewer for acknowledging the effectiveness and efficiency of our method. We would provide the following response to your comments.
>
> **Q:  The settings of parameters are underspecified or subjectively determined. The training/evaluation data are not widely available.**
>
> We would kindly remind the reviewer that we provide the detailed training and evaluation hyperparameters in Section 4.1.1, Section 4.2.1, and Appendix B.2. We use the fixed learning rate 0.0001 for continual training and search for the best learning rate for each method during the fine-tuning. For the other hyperparameters, we set them as standard values following the prior work. Please see the mentioned sections for more details. For the datasets, we use the publicly available datasets (ArXiv, PubMed, SCROLLS) to evaluate model performance on NLM and NLG tasks.

---

### Meta-Review · Area_Chair_5FRx · 2023-09-18

**Recommendation:** 3

**Metareview:**

Summary of the paper: The core idea presented in this paper is that for long-range language modeling and summarization, it is more critical to capture long-range dependencies early in the computation by using full attention in the bottom layers and sparse attention in the latter layers.

Pros:
- The simplicity of the idea makes it easy to replicate
- Their proposed setup works with pre-trained model

Cons:
- **Time and memory experiment**
Efficient transformers are useful because they reduce the compute and time complexity of transformers from quadratic in the sequence length to something less than quadratic in the sequence length. Since the proposed method uses fully attention as the first few layer the computational and memory requirement is $O(c_1L^2 + c_2Lk)$ where  L is the sequence length, k is the local attention block size, $c_1 + c_2$ is the total  number of transformer layers. So theoretically the proposed method is quadratic in sequence length.

Now, it is true that even if theoretically $L^2$ will dominate at large $L$ we could still think that the whole computation will be done much faster at lower $L$. First this is a drawback of the method not reflected in the experiments/analysis. Second, this needs to be demonstrated by measuring the computational time and memory required rather than “attention cost”. The reason is that compute time will be throttled by the lower layer, something that is not reflected in the “attention cost” metric. Moreover this tradeoff needs to be better analyzed in the paper.

- **Core idea of the paper and comparison to baselines**
Multiple Sparse Attention Transformers [1, 2, 3, 4, 5, 6] have been proposed over the past 4-5 years. The idea presented in the paper is combining a few fully attending layers at the bottom with local attention layers at the top. The closest idea to the paper is using few fully connected tokens (i.e. tokens that have full attention) in each layer. This has been explored  in *several* papers [4, 5, 6] along with an analysis of their compute and memory requirement [7]. Given the relationship of this paper with such previous work, the lack of comparison to efficient transformers, which are similar in principle, is another major concern for this paper.

---

### Decision · Program_Chairs · 2023-10-07

**Decision:**

Accept-Findings

**Comment:**

Summary of the paper: The core idea presented in this paper is that for long-range language modeling and summarization, it is more critical to capture long-range dependencies early in the computation by using full attention in the bottom layers and sparse attention in the latter layers.

Pros:
- The simplicity of the idea makes it easy to replicate
- Their proposed setup works with pre-trained model

Cons:
- **Time and memory experiment**
Efficient transformers are useful because they reduce the compute and time complexity of transformers from quadratic in the sequence length to something less than quadratic in the sequence length. Since the proposed method uses fully attention as the first few layer the computational and memory requirement is $O(c_1L^2 + c_2Lk)$ where  L is the sequence length, k is the local attention block size, $c_1 + c_2$ is the total  number of transformer layers. So theoretically the proposed method is quadratic in sequence length.

Now, it is true that even if theoretically $L^2$ will dominate at large $L$ we could still think that the whole computation will be done much faster at lower $L$. First this is a drawback of the method not reflected in the experiments/analysis. Second, this needs to be demonstrated by measuring the computational time and memory required rather than “attention cost”. The reason is that compute time will be throttled by the lower layer, something that is not reflected in the “attention cost” metric. Moreover this tradeoff needs to be better analyzed in the paper.

- **Core idea of the paper and comparison to baselines**
Multiple Sparse Attention Transformers [1, 2, 3, 4, 5, 6] have been proposed over the past 4-5 years. The idea presented in the paper is combining a few fully attending layers at the bottom with local attention layers at the top. The closest idea to the paper is using few fully connected tokens (i.e. tokens that have full attention) in each layer. This has been explored  in *several* papers [4, 5, 6] along with an analysis of their compute and memory requirement [7]. Given the relationship of this paper with such previous work, the lack of comparison to efficient transformers, which are similar in principle, is another major concern for this paper.